# HMP-S7 Is a Novel Anti-Leukemic Peptide Discovered from Human Milk

**DOI:** 10.3390/biomedicines9080981

**Published:** 2021-08-09

**Authors:** Wararat Chiangjong, Jirawan Panachan, Thitinee Vanichapol, Nutkridta Pongsakul, Pongpak Pongphitcha, Teerapong Siriboonpiputtana, Tassanee Lerksuthirat, Pracha Nuntnarumit, Sarayut Supapannachart, Chantragan Srisomsap, Jisnuson Svasti, Suradej Hongeng, Somchai Chutipongtanate

**Affiliations:** 1Pediatric Translational Research Unit, Department of Pediatrics, Faculty of Medicine Ramathibodi Hospital, Mahidol University, Bangkok 10400, Thailand; wararat.chi@mahidol.ac.th (W.C.); nutkridta.pon@mahidol.ac.th (N.P.); 2Department of Pediatrics, Division of Hematology and Oncology, Faculty of Medicine Ramathibodi Hospital, Mahidol University, Bangkok 10400, Thailand; jirawan.pan@mahidol.ac.th (J.P.); thitinee@gmail.com (T.V.); pongpak.cha@mahidol.ac.th (P.P.); suradej.hon@mahidol.ac.th (S.H.); 3Department of Pathology, Faculty of Medicine Ramathibodi Hospital, Mahidol University, Bangkok 10400, Thailand; teerapong.sir@mahidol.ac.th; 4Research Center, Faculty of Medicine Ramathibodi Hospital, Mahidol University, Bangkok 10400, Thailand; tassanee.ler@mahidol.ac.th; 5Division of Neonatology, Faculty of Medicine Ramathibodi Hospital, Mahidol University, Bangkok 10400, Thailand; pracha.nun@mahidol.ac.th (P.N.); sarayut.sup@mahidol.ac.th (S.S.); 6Laboratory of Biochemistry, Chulabhorn Research Institute, Bangkok 10210, Thailand; chantragan@cri.or.th (C.S.); jisnuson@cri.or.th (J.S.); 7Applied Biological Sciences Program, Chulabhorn Graduate Institute, Bangkok 10210, Thailand; 8Department of Clinical Epidemiology and Biostatistics, Faculty of Medicine Ramathibodi Hospital, Mahidol University, Bangkok 10400, Thailand; 9Faculty of Medicine Ramathibodi Hospital, Chakri Naruebodindra Medical Institute, Mahidol University, Samut Prakan 10540, Thailand

**Keywords:** anticancer peptide, drug discovery, human milk, leukemia, machine learning, mass spectrometry, peptidomics

## Abstract

Chemotherapy in childhood leukemia is associated with late morbidity in leukemic survivors, while certain patient subsets are relatively resistant to standard chemotherapy. It is therefore important to identify new agents with sensitivity and selectivity towards leukemic cells, while having less systemic toxicity. Peptide-based therapeutics has gained a great deal of attention during the last few years. Here, we used an integrative workflow combining mass spectrometric peptide library construction, in silico anticancer peptide screening, and in vitro leukemic cell studies to discover a novel anti-leukemic peptide having 3+ charges and an alpha helical structure, namely HMP-S7, from human breast milk. HMP-S7 showed cytotoxic activity against four distinct leukemic cell lines in a dose-dependent manner but had no effect on solid malignancies or representative normal cells. HMP-S7 induced leukemic cell death by penetrating the plasma membrane to enter the cytoplasm and cause the leakage of lactate dehydrogenase, thus acting in a membranolytic manner. Importantly, HMP-S7 exhibited anti-leukemic effects against patient-derived leukemic cells ex vivo. In conclusion, HMP-S7 is a selective anti-leukemic peptide with promise, which requires further validation in preclinical and clinical studies.

## 1. Introduction

Cancer is a significant cause of death in children and adolescents during the last few years in Asia, Central and South America, Northwest Africa, and the Middle East [1]. Hematologic malignancies, particularly acute lymphocytic leukemia (ALL), are predominant, accounting for 30% of childhood cancers [2]. The optimization of chemotherapeutic regimens during the past decades has resulted in more than a remission rate of 90% in childhood ALL [3]. However, the burden of late morbidity due to chemotherapeutic treatments, which occur in two-thirds of pediatric cases [4], has become important considerations, as the number of long-term leukemic survivors increases [5]. Moreover, specific subsets of pediatric ALL are relatively resistant to standard chemotherapy and have high risk of relapse [6]. These challenges stress the need to further improve current treatment, while identifying new agents with sensitivity and selectivity toward leukemic cells with little to no toxicity to normal cells.

Peptide-based drugs or anticancer peptides can provide a new strategy for cancer treatment. Although the exact mechanisms and selectivity criteria have yet to be elucidated, anticancer peptides may have oncolytic effects depending on peptide characteristics and target membrane properties in determining selectivity and toxicity [7]. Cancer cells have highly negative transmembrane potential from anionic molecules on the surface, greater membrane fluidity, and more abundant microvilli (increasing outer surface area). In contrast, normal cells are electrically neutral [8,9,10,11]. The negative charges on the cancer cell membrane attract positively charged peptides which will disturb membrane stability, causing the loss of electrolytes and cell death [8], while the high cholesterol content of the normal cell membrane can protect cell fluidity and block cationic peptide entry [12]. Common strategies for anticancer peptide discovery include the following: (a) activity-guided purification from biological/natural products [13]; (b) examination of antimicrobial peptides for cancer sensitivity [14,15]. The former approach tends to deliver positive results but is associated with labor-intensive and time-consuming processes. The latter strategy is relatively cost- and time-effective but still tends to be limited to antimicrobial peptides known a priori. The investigation of anticancer peptides has progressed slowly in the past decades, indicating gaps for improvement, particularly on the selection of peptide sources, screening methods, and downstream analyses.

Human milk is a promising source of therapeutic peptides. Bioactive milk peptides are released from source proteins by enzymatic hydrolysis, fermentation with a proteolytic starter culture, and proteolysis with proteolytic microorganisms [16,17]. Although human milk has rarely been investigated for anticancer activities, studies have shown human-milk-derived peptides could exhibit various biological effects such as antimicrobial [18,19] and immunomodulatory activities [20]. Human-milk-derived beta-casein fragments have been studied for immunomodulation, antibacterial, antioxidant, opioid agonist, antihypertensive activities, as well as cell proliferation of human preadipocytes [21]. These bioactive peptides have different amino acid compositions and sequences, and the peptide length can vary from two to twenty amino acid residues [22]. It was anticipated that human milk also contains anticancer peptides, as found with bovine milk. A hexapeptide Pro-Gly-Pro-Ile-Pro-Asn (PGPIPN; residues 63–68) from bovine β-casein can inhibit the invasion and migration of human ovarian cancer cells [23]. Anticancer Fusion Peptide (ACFP), an anticancer fusion peptide derived from bovine β-casein and lactoferrin, can inhibit viability and promote apoptosis in primary ovarian cancer cells [24]. Interestingly, breastfeeding for six months or longer is associated with a 19% lower risk of all childhood leukemia than shorter breastfeeding or none [25,26]. Identifying novel antileukemic peptides from human milk is important for the future development of nonallergic and nontoxic peptide-based drugs to improve childhood leukemia therapy.

This study aimed to discover a novel antileukemic peptide from human milk. We applied a robust workflow integrating the strength of liquid chromatography-tandem mass spectrometry to generate a library of naturally occurring human milk peptides, in silico screening based on physicochemical, structural, and predictive anticancer properties (ACPs), using machine learning algorithms. Potential candidates were selected for functional studies of synthetic peptides to determine antileukemic activity and identify the mode of action. By this strategy, a novel antileukemic peptide was identified from human milk as the main outcome, and its antileukemic activity was successfully validated against four distinct leukemic cell lines in vitro, as well as three patient-derived leukemic cells ex vivo. 

## 2. Materials and Methods

### 2.1. Experimental Design

This study aimed to identify a novel antileukemic peptide from human milk. An integrative approach was established by combining the strengths of mass spectrometric-peptide identification for library construction, the in silico screening of peptide library for prioritizing peptide candidates, and in vitro experimental validation for antileukemic activities. The activity of a newly discovered antileukemic peptide against patient-derived leukemic cells ex vivo was finally examined. All subjects gave their informed consents for inclusion, before they participated in the study. The study was conducted in accordance with the Declaration of Helsinki, and the protocol was approved by the Human Research Ethics Committee, Faculty of Medicine Ramathibodi Hospital, Mahidol University (Protocol ID 11-60-13, No. MURA2017/760; with an approval of amendment on 3 May 2021).

### 2.2. Human Milk Collection

Human milk (60 mL) from 10 healthy volunteer mothers of full-term infants, whose blood tests had no infection of HIV, Hepatitis B or C viruses, and Syphilis, was collected by a breast pump and stored in a freezer (−20 °C) as per the regulation of Ramathibodi Human Milk Bank (RHMB) before transferring to the laboratory [27].

### 2.3. Peptide Isolation and Fractionation

Three pooled milk specimens were produced from 3, 3, and 4 individual specimens (Appendix A) before further processing. Twenty milliliters of pooled milk specimens were centrifuged at 1500× *g* and 4 °C for 10 min and then at 5000× *g* for 30 min twice to remove cells and lipids. The collected supernatant was centrifuged at 12,000× *g* and 4 °C for 1 h, at 32,000× *g* for 1 h, and finally, at 200,000× *g* for 1 h to remove extracellular vesicles, including microvesicles and exosomes. Thereafter, peptides (<3 kDa) were separated from high-molecular-size proteins by a 3 kDa cutoff ultrafiltration column (Amicon^®^, Merck Millipore Ltd., Cork, Ireland). The crude peptides (<3 kDa) in the flow-through fraction were injected into the C18 solid-phase extraction (SPE) column (Waters, Waters Corporation, MA, USA). The bound peptides were then stepwise elution with 1 mL each of 15%, 20%, 25%, 30%, 35%, 40%, 45%, 50%, 55%, and 80% acetonitrile (ACN; Sigma-Aldrich, St. Louis, MO, USA), respectively. The eluted peptides were dried using a SpeedVac concentrator (Labconco, Labconco Corporation, MI, USA). The dried peptides were then resuspended in deionized (d*I*) water for peptide estimation using Bradford’s assay (Bio-rad, Hercules, CA, USA) [28].

### 2.4. Synthetic Peptides

Nine synthetic peptides (purity: >98%) (Appendix A) were custom-ordered from GL Biochem (GL Biochem (Shanghai) Ltd., Shanghai, China) and resuspended in an appropriate culture medium to the final concentration before use. These included the following: HMP-S1, NH2-TIESLSSSEESITEYK-COOH; HMP-S2, NH2-ADSGEGDFLAEGGGVR-COOH; HMP-S3, NH2-PPPPPPPPP-COOH; HMP-S4, NH2-APGPP-COOH; HMP-S5, NH2-VSLISAEVPLGR-COOH; BMP-S6 (the positive control [29]), NH2-FKCRRWQWRMKKLGAPSITCVR-COOH; HMP-S7, NH2-SFIPRAKSTWLNNIKLL-COOH; HMP-S8, NH2-GRATLVQDGIAKGRVA-COOH; HMP-S9, NH2-LPIPQQVVPYPQRAVPVQ-COOH.

### 2.5. Cell Culture

Cell lines were obtained from the American Type Culture Collection (ATCC). Jurkat (ATCC^®^ TIB-152™), RS4;11 (ATCC^®^ CRL-1873™), and Raji (ATCC^®^ CCL-86™) cells were maintained in an RPMI-1640 medium (Gibco, Thermo Fisher Scientific, Waltham, MA, USA) supplemented with 10% fetal bovine serum (FBS) (Gibco) and 1× penicillin/streptomycin (Gibco) in 5% CO_2_ at 37 °C, while Sup-B15 (ATCC^®^ CRL-1929™) cells were cultured in Iscove’s modified Dulbecco’s medium (IMDM) supplemented with 0.05 mM 2-mercaptoethanol (Sigma), 20% FBS (Gibco), and 1× penicillin/streptomycin (Gibco). SH-SY5Y (ATCC^®^ CRL-2266™), MDA-MB-231 (ATCC^®^ HTB-26™), and A549 (ATCC^®^ CCL-185™) cell lines were grown in Dulbecco’s Modified Eagle’s Medium (DMEM)-high glucose (Gibco) supplemented with 10% FBS and 1× penicillin/streptomycin in 5% CO_2_ at 37 °C. HT-29 cells (ATCC^®^ HTB-38™) and HepG2 (ATCC^®^ HB-8065™) were grown in DMEM-F12 (Gibco) supplemented with 10% FBS and 1× penicillin/streptomycin in 5% CO_2_ at 37 °C. For representative normal cells, HEK293T human embryonic kidney cells (ATCC^®^ CRL-3216™) were maintained in DMEM with high glucose (Gibco) supplemented with 10% FBS and 1× penicillin/streptomycin in 5% CO_2_ at 37 °C. Human peripheral blood mononuclear cells (PBMCs) were prepared from 10 mL EDTA blood for T cell cultivation. The whole blood was diluted in PBS at a volume ratio of 1:1 and then carefully layered into a Ficoll-Paque solution (Robbins Scientific Cooperation, Norway) with a volume ratio of 2:1 (diluted blood:Ficoll-Paque solution). The tube containing the layered solution was centrifuged at 400 × *g* and 20 °C for 35 min with no break. One million of PBMCs were cultured in OKT3- and CD28-coated plates. Briefly, 1 µg of OKT3 and CD28 each was added into 1 mL PBS in a 24-well plate and incubated at room temperature for 2 h. The coated well was washed with PBS once before adding PBMCs. The PBMCs in the coated well were cultured in RPMI-1640 supplemented with 100 U/mL IL-2, 10% FBS, and 1× penicillin/streptomycin in the presence of OKT3 and CD28 for 3 days at 37 °C under a humidified atmosphere with 5% CO_2_ before use. 

### 2.6. Human Milk Peptide Identification by Mass Spectrometry

Dried human milk peptides (2 µg) were resuspended in 0.1% formic acid and centrifuged at 14,000 rpm for 30 min. The supernatant was collected and injected into the C18 column (75 µm i.d. × 100 mm) by using Easy-nLC (Thermo Fisher Scientific) to desalt and concentrate. The peptides were separated with a gradient of 5–45% ACN/0.1% formic acid for 30 min at a flow rate of 300 nl/min. The isolated peptides were identified by the amaZon speed ETD ion trap mass spectrometer (Bruker Daltoniks, Billerica, MA, USA). Peptide sequences and identifications were interpreted using in-house MASCOT software version 2.4.0 with the SwissProt database, against Homo sapiens, with no fixed and no variable modifications, no enzymatic digestion with no missed cleavage allowed, monoisotopic, ±1.2 Da for peptide tolerance, ±0.6 Da for fragment ion tolerance, and 2+ and 3+ charge states for ESI-TRAP instrument. Identified peptides had ion scores higher than 20 (significance threshold at *p* < 0.05).

### 2.7. In Silico Anticancer Peptide Screening

The physicochemical properties of all identified human milk peptides were predicted by using the PepDraw tool; available online: http://www.tulane.edu/~biochem/WW/PepDraw/ (accessed on 21 January 2020), including peptide length, mass, net charge, isoelectric point (p*I*), and hydrophobicity. Peptide structures were predicted using the PEP-FOLD3 De novo peptide structure prediction tool; available online: https://mobyle.rpbs.univ-paris-diderot.fr/cgi-bin/portal.py#forms::PEP-FOLD3 (accessed on 17 January 2020) [30,31,32]. ACPs were predicted by four web-based machine learning programs as following: (i) ACPred-FL; available online: http://server.malab.cn/ACPred-FL/# (accessed on 21 January 2020), a sequence base predictor for identifying anticancer peptides by using a classification mode at a confidence of 0.5 as a default setting [33]; (ii) AntiCP 2.0; available online: https://webs.iiitd.edu.in/raghava/anticp2/index.html (accessed on 3 March 2021), an ensemble tree classifier based on amino acid composition [34]; (iii) MLACP; available online: www.thegleelab.org/MLACP.html (accessed on 21 January 2020), a random forest-based prediction of anticancer peptides [35]; and (iv) mACPpred; available online: http://www.thegleelab.org/mACPpred/ (accessed on 21 January 2020), a Support Vector Machine-Based Meta-Predictor [36].

### 2.8. Cytotoxicity Assay

For cell treatment, the fractionated milk peptides (10 µg/reaction) or the synthetic peptides (varied concentrations as indicated) were resuspended in 100 µL of the culture media with appropriate supplements. For floating cells, the peptide solution was mixed with cell suspension (10,000 cells/10 µL/well) in a well of a 96-well flat-bottomed plate. For adherent cells, 10,000 cells were seeded into a 96-well plate (flat bottom) and cultured until they reached 80% confluence before adding 100 µL of the peptide solution into the well containing 80% FHs 74 Int fetal intestinal cell confluence (3 technical replicates). The peptide-treated cells were incubated for 24, 48, or 72 h in a 5% CO_2_ incubator at 37 °C with humidity as indicated. Cell viability was measured by trypan blue exclusion or WST-1 assays (Roche Diagnostics GmbH, Mannheim, Germany).

### 2.9. Half-Maximal Inhibitory Concentration (IC_50_) by Trypan Blue Assay

Four distinct leukemic cell lines, i.e., Jurkat, Raji, RS4;11, and Sup-B15 cell lines, were cultured in a medium containing HMP-S7 in varying concentrations (0, 6.25, 12.5, 25, 50, 100, 200, and 400 µM) in 96-well flat-bottomed plates for 24 h (1 × 10^4^ cells/100 µL/well). The percentage of cell death was estimated by employing the trypan blue exclusion assay and calculated as: cell death number/total cell number × 100%. IC_50_ calculation used the linear (y = ax + c) or parabolic (y = ax^2^ + bx + c) equation for y = 50 value and x value = IC_50_ concentration. The IC_50_ concentrations of HMP-S7 to the 4 leukemic cell lines were reported as mean ± SD.

### 2.10. Soft Agar Assay for Colony Formation

Base agars (1.5 mL of 0.5% agar containing 1× RPMI supplemented with 10% FBS and 1× penicillin/streptomycin) were plated on each well of a 6-well plate and set aside for 5 min to allow the agars to solidify. Jurkat cells were treated with 100, 200, and 400 µM HMP-S7 and 200 µM BMP-S6 in a 96-well plate (3 technical replicates; 100 µL/10,000 cells/well) for 24 h. After 24 h, the treated/untreated cells in the 96-well plate were thoroughly mixed and 20 µL were taken to mix in 1 mL of a top agar solution (0.3% agarose containing 1× RPMI supplement with 10% FBS and 1× penicillin/streptomycin). The cell suspension was plated on top of the base agars (3 biological replicates) and then the agarose allowed solidifying. The medium (0.5 mL) was added on top of the agars to prevent the agars from drying. The plate was incubated at 37 °C in a humidified incubator for 20 days with the medium being added twice a week. The agars were stained with 1 mL of 0.005% crystal violet in 20% ethanol for 1 h and destained with 20% ethanol overnight, and then the colonies were counted under a stereomicroscope SZ61 (Olympus Corporation, Tokyo, Japan).

### 2.11. Flow Cytometric Cell Death Assay

Four leukemic cell lines (1 × 10^5^ cells/500 µL/well/cell type) were cultured in a 24-well plate in a medium containing HMP-S7 with IC_50_ and 2 × IC_50_ concentrations and untreated condition for 24 h (biological triplicate). The untreated cells were divided into 3 tubes representing unstained cells, cells stained with fluorescein isothiocyanate (FITC)-tagged annexin V (no propidium iodide (PI)), and cells stained with PI (no FITC tagged annexin V) to set up compensation and quadrants. The untreated and treated cells were collected and washed twice with cold PBS. Thereafter, the cells were resuspended with 100 µL of a 1× binding buffer, and then, 5 µL of FITC tagged annexin V and 5 µL of PI were added (BD Biosciences, CA, USA). The cell suspensions were gently mixed and incubated at room temperature for 15 min in dark. Then, 400 µL of the 1× binding buffer were added to each tube, and then the cells were analyzed by using flow cytometry.

### 2.12. Lactate Dehydrogenase (LDH) Release Assay

RS4;11 leukemic cell lines (2 × 10^5^ cells/1 mL/well/cell type) were cultured in 24-well plates in a complete medium containing HMP-S7 at IC_50_ and 2 × IC_50_ concentrations, without HMP-S7 conditions (non-treatment) and 0.5% Triton X-100 for 24 h incubation (biological triplication). The cell suspension was collected and centrifuged at 200× *g* for 5 min at room temperature. Then, 1 mL of the supernatant was collected, and LDH enzyme was measured from the conversion of lactate and NAD^+^ to pyruvate and NADH. NADH products were detected at the absorbance of 340 nm by using Abbott Architect C16000 clinical chemistry analyzer (Holliston, MA, USA) at the Clinical Chemistry Unit, Department of Pathology, Ramathibodi Hospital, Thailand. All reagents, including R1 (381 mM diethanolamine, 76 mM L-lactate and sodium azide (<0.1%)) and R2 (30.8 mM β-NAD) were obtained from Abbott Diagnostics (Abbott Laboratories, Lake Forest, IL, USA). The LDH level (U/L) of the culture media was subtracted with the background of the fresh medium and reported as mean ± SD.

### 2.13. FITC Tagged HMP-S7 Internalized into Leukemic Cells

Each well containing 2 × 10^4^ Jurkat cells/100 µL in the 96-well plate was treated with HMP-S7 tagged with/without FITC at IC_50_ for 24 h. Thereafter, the treated and untreated Jurkat cells were washed once with PBS before staining with PI cell stain kit (Invitrogen, Waltham, MA, USA) and Hoechst 33342 (Cell Signaling Technology, Inc., Danvers, MA, USA) for 30 min at room temperature in dark. After incubation, the cells were centrifuged at 200× *g* for 5 min, and excess dye was removed before washing twice with PBS. The cell pellet was mixed with 20% glycerol/PBS and mounted on a glass slide for confocal microscopy (Nikon Instruments, Inc., Melville, NY, USA).

### 2.14. Ex Vivo Leukemic Cytotoxicity Study

Patient-derived leukemic cells were obtained from Ramathibodi Tumor Biobanking. These cells were isolated from bone marrow-aspirated samples obtained from three B cell-acute lymphoblastic leukemia (B-ALL) patients by Ficoll-Paque PLUS density gradient centrifugation, washed by sterile PBS, resuspended in FBS/10% DMSO and then stored in liquid nitrogen until use. Demographic and clinical data were curated from the Electronic Medical Record and the patient’s chart. Patient-derived leukemic cells were thawed in a water bath at 37 °C for 5 min, washed with sterile PBS, resuspended and maintained ex vivo in RPMI supplemented with 20% FBS and 1× penicillin/streptomycin for 7 days. Thereafter, patient-derived leukemic cells were collected by centrifugation at 300× *g* for 5 min at room temperature and counted by LUNA-II automated cell counter (Logos Biosystems, Inc., Gyeonggi-do, South Korea). Each specimen was stained by Wright staining and evaluated under a light microscope to ensure that greater than 90% morphologically recognizable malignant lymphoblasts were detected before further analysis. Approximately 5 × 104 leukemic cells were then treated with 200 µM BMP-S6, 200 and 400 µM HMP-S7 peptides, or not treated (4 replicates per condition) in a 24-well plate. After 3 days of treatment, all cells were collected and washed twice with sterile PBS. The cell pellets were resuspended in a 1× binding buffer and then stained with FITC Annexin V/PI apoptosis detection assay (BD Biosciences, CA, USA). The % cell death was determined by using flow cytometry (BD FACSVerse, BD Biosciences).

### 2.15. Statistical Analysis

The number and percentage of dead or live cells, and IC_50_, were calculated and statistically tested using Student’s t-test with statistical significance at *p* < 0.05.

## 3. Results

To separate human milk peptides and test their cytotoxicity to leukemic cells, 10 healthy mothers aged between 25 and 36 years old (32.3 ± 3.27 years old) donated breast milk at once for 6–129 days (63.8 ± 38.97 days) after delivery (details in Appendix A). After multiple steps of centrifugation to remove cells, lipids, and extracellular vesicles, ultrafiltration (3 kDa cutoff) was used to remove proteins and the small molecules (<3 kDa) were subjected to C18 SPE. The C18-bound peptides were eluted by various ACN concentrations and tested for their cytotoxicity. The conceptual framework of this study is illustrated in Figure 1.

### 3.1. Most Human Milk Peptide Fractions Had Cytotoxic Effects on Leukemic and Normal Cell Lines

After C18-bound peptide elution, the crude milk peptide fraction was tested to observe the effects on the cell viability of Jurkat (T lymphoblastic leukemia) and FHs74Int cells (the representative normal intestinal epithelium) (Figure 2a). The crude milk peptides showed a potent cytotoxic effect on Jurkat but had less impact on FHs74Int cells. The coarse milk peptide fraction was then fractionated by C18 SPE of the stepwise ACN elution, with the chromatogram showing amounts of peptides at each % ACN (Figure 2b). Peptides eluted with 15–80% ACN were used to treat Jurkat leukemic cells and FHs74Int normal cells, and cell survival was observed as shown in Figure 2c. The relatively low peptide content in the 15–30% ACN eluted fractions could decrease Jurkat cell survival more than the fraction with the highest peptide content (45% ACN eluted peptide fraction). This indicated that the elimination of Jurkat leukemic cells depended on the specific peptide sequence with hydrophilic properties being more than the peptide amount.

### 3.2. Peptide Identification, Library Construction and In Silico Anticancer Peptide Screening

Eleven fractions (from one crude and 10 stepwise ACN eluates) of human milk peptides were identified by liquid chromatography-tandem mass spectrometry (LC-MS/MS). A total of 142 naturally occurring human milk peptides with unique sequences were collected into the peptide library, and the properties of each peptide are shown in Appendix A. The distributions of all identified peptides by fractions, peptide length, p*I*, and the net charge are shown in Figure 3a (upper panel). Overall, most identified naturally occurring milk peptides had fewer than 20 amino acids in length and contained zero or anionic net charge inclined towards acid properties. The most frequently detected pattern was found to be natural peptides with proline-rich sequences (Appendix A). Proline-rich peptides may play important roles in the biological effects of human milk. For example, a ligand containing a proline-rich sequence or a single proline residue may be involved in protein–protein interaction [37]. Antimicrobial peptides with proline-rich sequences can kill microorganisms by interacting with 70S ribosome and disrupting protein synthesis [38].

The in silico screening of the naturally occurring human milk peptide library was performed using a robust workflow. Firstly, we screened the library by physicochemical properties, i.e., length, *p*I, and net charge (as shown in Figure 3a and Appendix A), since most of the known anticancer peptides are small cationic peptides (commonly 5–30 amino acids in length with net charges from +2 to +9) that bind to the negative charge of phosphatidylserine and sialic acid on the cancer cell membrane [12,39,40]. Secondly, the secondary structures of the peptides were predicted by the PEP-FOLD3 webserver (Figure 3b and Appendix A). Peptides can form the alpha helix, the beta-sheet, and random coils [41]. However, most of the known anticancer peptides share the alpha helix structure [42]. Thirdly, the identified peptides were predicted for ACPs using machine learning prediction software. Since ACP prediction software can provide different results depending on algorithms and training datasets, four kinds of online-accessible software were applied in this study, including ACPpred-FL [33], antiCP 2.0 [34], MLACP [35], and mACPpred [36] (Figure 3c and Appendix A). From the data shown in Appendix A, eight human milk peptides were selected as having ACP potential, based on net charge, secondary structure, and predicted anticancer property, and used for further analysis. The sequences and properties of these eight selected human milk peptides (HMP-S1, HMP-S2, HMP-S3, HMP-S4, HMP-S5, HMP-S7, HMP-S8, and HMP-S9) are shown in Appendix A, together with those of a bovine milk peptide BMP-S6 used as a positive ACP control.

### 3.3. HMP-S7 Killed Leukemic Cells but Not Normal Cells or Solid Tumor Cell Lines

Figure 4a shows the eight selected human-milk-derived peptides and a known bovine-milk-derived anticancer peptide, which were synthesized to screen for cancer cytotoxicity by using trypan blue staining. The results showed that HMP-S7, the most highly charged cationic peptide (net charge: +3) with an α-helical structure, had higher inhibitory activity than the other human-milk-derived peptides towards four leukemic cell lines, including Jurkat, Raji, RS4;11, and Sup-B15 cells (Figure 4b). To study the selectivity of HMP-S7, the effect of HMP-S7 was further observed on normal cells, namely T cells and HEK293T embryonic kidney cells, as well as on solid tumor cell lines, namely SH-SY5Y neuroblastoma, HepG2 hepatoblastoma, A549 lung adenocarcinoma, MDA-MB-231 breast cancer, and HT-29 colorectal adenocarcinoma (Figure 4c). Compared to BMP-S6, a known anticancer peptide, HMP-S7, showed less cytotoxic activity towards normal cells and most cancer cells. This suggested that HMP-S7 was more selective to leukemic cells than BMP-S6. Although HMP-S7 showed no cytotoxicity to normal T cells, it had a mild cytotoxic effect on HEK293T cells. Based on these activity profiles, HMP-S7 was chosen to further studies to determine its IC_50_ against four leukemic cell lines (Figure 5a). The results demonstrated the antileukemic activity of HMP-S7 was dose-dependent, with the IC_50_ ranging from 89.2 to 186.3 μM depending on the leukemic cell tested. Furthermore, the inhibitory activity of the HMP-S7-treated Jurkat leukemic cell lines was confirmed by the colony-forming assay using soft agar, as shown in Figure 5b. Since HMP-S7 could induce leukemic cell death; thereafter, the mechanism of cell death was further investigated.

### 3.4. Mechanisms of HMP-S7-Induced Leukemic Cell Death

To elucidate whether HMP-S7 can act as a membranolytic peptide towards leukemic cells, Jurkat cells were treated with HMP-S7, in untagged form or tagged with FITC at the IC_50_ for 24 h. By confocal microscopy, the PI staining of cytoplasmic RNA was observed in the untagged HMP-7 and the positive control Triton X-100 (Figure 6a). The FITC-tagged HMP-S7 was located in the cytoplasm as with the PI stain of cytoplasmic RNA. The flow cytometry showed that HMP-S7 could accumulate more in the cytoplasm with increasing concentrations from IC_50_ to 2 × IC_50_ (Figure 6b,c). Using the LDH release assay, intracellular LDH leakage into the cell culture supernatant was evident at 24 h in HMP-S7-treated RS4;11 cells (at IC_50_ and 2 × IC_50_) and Triton X-100-treated cells (Figure 6d). Finally, HMP-S7-induced leukemic cell death was confirmed by flow cytometry using Annexin V/PI co-staining. The number of apoptotic cells was significantly increased compared to under untreated condition, while a dose-response relationship was observed in four leukemic cell lines with IC_50_ and 2 × IC_50_ of HMP-S7 treatment (Figure 6e,f). Taken together, our findings suggested that HMP-S7 attacked leukemic cells to form micropores in the cell membranes and penetrated into the cytoplasm to perturb leukemic cells, leading to cell death.

### 3.5. Ex Vivo Leukemic Cytotoxicity of HMP-S7

Finally, we studied whether HMP-S7 can exhibit antileukemic effects against patient-derived leukemic cells ex vivo. Bone-marrow-derived mononuclear cells were isolated from three patients with B-ALL by Ficoll density gradient centrifugation (demographic data as shown in Figure 7a). The percentages of leukemic cells in bone marrow specimens were 84.5–92.5% (Figure 7a). Nonetheless, greater than 90% morphologically recognizable malignant lymphoblasts were achieved after mononuclear cell isolation and ex vivo culturing (data not shown). These cells were assigned as patient-derived leukemic cells for ex vivo leukemic cytotoxic assay using Annexin V/PI co-staining. Interestingly, HMP-S7 (200 and 400 µM) exhibited antileukemic activity against ex vivo leukemic cells isolated from three independent patients in a dose-dependent manner (Figure 7b,c). BMP-S6 (200 µM), which demonstrated antcancer activity against multiple cancers including leukemia cell lines (Figure 4 and Figure 5), was included for comparison. However, BMP-S6 had a marginal-to-no cytotoxic effect against patient-derived leukemic cells ex vivo (Figure 7b,c). This finding supports further development of HMP-S7 as an antileukemic peptide for preclinical and clinical studies.

## 4. Discussion

This study aimed to search for a novel ACP from naturally occurring human milk peptides. Several reasons suggest that human milk would be a good source for this. First, there is evidence that human breast milk can significantly reduce leukemia in children with more than 6 months of breastfeeding [25,26]. Secondly, human alpha-lactalbumin made lethal to tumor cells (HAMLET), a protein–lipid complex that induces apoptosis-like death in tumor cells, was discovered from a human milk protein [43]. Alpha1H (the alpha1 domain of α-lactalbumin in complex with oleic acid), which was further developed from HAMLET, has now entered the First-in-Human trial in patients with bladder cancer (clinicaltrials.gov identifier NCT03560479) [44]. Thirdly, ACPs have never been explored from human milk. Fourthly, recent advances in mass spectrometric-based proteomics allow high-throughput peptide identification from human milk, and in silico ACP prediction algorithms based on peptide sequences have improved in recent years.

Therefore, this study developed an integrative workflow for ACP discovery by combining the strengths of mass spectrometry, in silico screening, and experimental validation. This strategy replaces labor-intensive and time-consuming processes of the activity-guided purification by applying the mass spectrometric peptide identification of fractionated human milk peptides and the compilation of the results into a peptide library of naturally occurring human milk peptides. Peptide amino acid sequences were subsequently subjected to in silico screening to shorten the time to discovery and lessen the cost of large-scale experiments. Finally, the prioritized peptide candidates could be validated for antileukemic activities using in vitro cellular studies.

To prioritize the peptides for experimental screening and validation, we hypothesized that the consensus results from the physicochemical property (positive charge), secondary peptide structure (alpha helix) and ACP-predicted results from different machine learning models would provide the best chance for the identification of ACP from the peptide library. This hypothesis was grounded on previous evidence as following: (i) the net positively charged peptides are attracted to the net negatively charged cancer cell membranes [45]; (ii) most known oncolytic peptides share an alpha helical structure [42]; (iii) different machine learning models were trained and tested upon various datasets of known ACPs [33,34,35,36], so these models have varied predictive performance against the new unknown peptide dataset. More positive predictions would provide more confidence in the predicted candidates. Since it was unknown at the initial stage of this study whether this integrative approach would be successful, we therefore explored each of the preferred ACPs. For this reason, eight selected peptide candidates were identified and screened for antileukemic effects (Figure 3 and Appendix A). BMP-S6, the positive control of the experimental validation, met all three criteria and showed an antileukemic effect with the trade-off of more toxicity toward normal cells. Six out of eight selected peptide candidates did not meet all three criteria and showed no activity in vitro. HMP-S7 and HMP-S8 were the top two candidates meeting all three criteria of the integrative workflow; nonetheless, only HMP-S7 had cytotoxic effects against leukemic cell lines in vitro (Figure 4, Figure 5 and Figure 6) and patient-derived leukemic cells ex vivo (Figure 7). From this observation, we learned that this integrative approach is more accurate than using a single preferred ACP to prioritize peptide candidates. When one peptide candidate met all three criteria, the difference in one positive charge (and perhaps two predictions from machine learning models) did matter for the antileukemic prediction (Figure 3). In future studies, this integrative approach of ACP screening can be applied to larger datasets, using either mass spectrometric-based or in silico generated peptide libraries, to speed up the discovery of ACPs against multiple cancer types.

Looking forward, HMP-S7 should be further validated in preclinical models. Further peptide modifications using HMP-S7 as a prototype, for example, amino acid substitutions [46], homing motif tagging [47], and PEGylation [48], may improve anticancer efficacy, tumor targeting, biocompatibility, and stability. Furthermore, the human milk peptide library can be expanded by further profiling relevant biological specimens or by in silico peptide generation from the proteins of interest. As mentioned for the in vitro study support, machine learning for ACP prediction might be further improved by a combination of ensemble models, physicochemical properties, and peptide secondary structures.

This study has limitations. First, it was expected that milk peptides should represent a more significant number of unique identities than those identified in this study. Loss of peptides during mass spectrometric detection due to neutral net charge may be one reason. Some milk peptides may also be lost and degraded during the fractionation process. Addressing these issues would improve the number of unique peptides identified by mass spectrometry in future studies. Secondly, given that the custom-made peptides are commercially available and ready for anticancer activity screening, the issues related to having a small peptide library because of experimental bottlenecks (specimen types, separation processes, and instrumental limits of detection) could be addressed by in silico generated peptide databases as mentioned above. Thirdly, our workflow is compatible with the native peptides, but this integrative workflow omitted peptides with post-translational modifications (such as glycosylated peptides) by default. Human-milk-derived glycopeptidomics could be important for therapeutic peptide investigations, once high-throughput data acquisition strategies and computational predictive tools are tailored for large-scale glycosylated peptide analysis. Lastly, it should be emphasized that the ex vivo antileukemic activity of HMP-S7 was performed on leukemic cells derived from patients with ALL. Future studies should consider evaluating the antileukemic effect of HMP-S7 in a broader context of hematologic malignancies, including myeloid leukemia and lymphoma.

## 5. Conclusions

This study applied an integrative workflow to discover HMP-S7 (NH_2_-SFIPRAKSTWLNNIKLL-COOH) as a novel antileukemic peptide derived from human breast milk. This antileukemic peptide is a cationic peptide with an alpha helical structure that selectively kills leukemic cell lines in vitro and exhibits its cytotoxicity against patient-derived leukemic cells ex vivo. Future research on peptide modification, together with the efficacy studies in preclinical animal models or early-phase clinical trials, is warranted to develop HMP-S7 as peptide-based cancer therapeutics in the future.

## Figures and Tables

**Figure 1 biomedicines-09-00981-f001:**
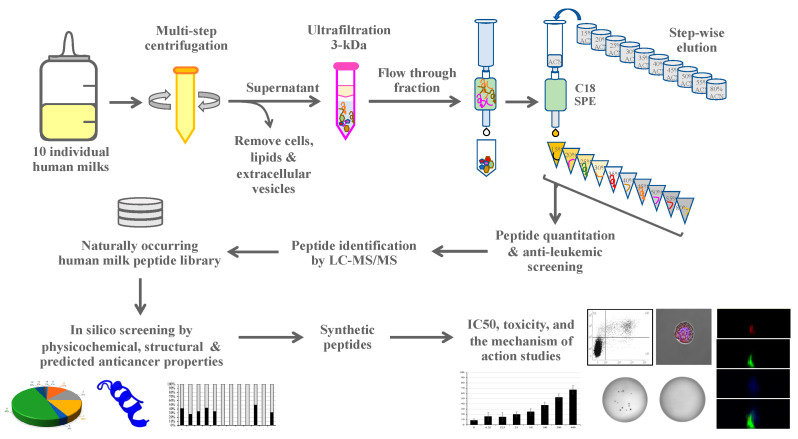
A conceptual framework of the integrative strategy for discovering a novel human-milk-derived antileukemic peptide. This strategy combines the strengths of mass spectrometry for high-throughput peptide identification, in silico screening for prioritizing peptide candidates, and experimental validation for antileukemic activities. Abbreviations: LC-MS/MS, liquid chromatography-tandem mass spectrometry; IC_50_, half-maximal inhibitory concentration; SPE, solid-phase extraction.

**Figure 2 biomedicines-09-00981-f002:**
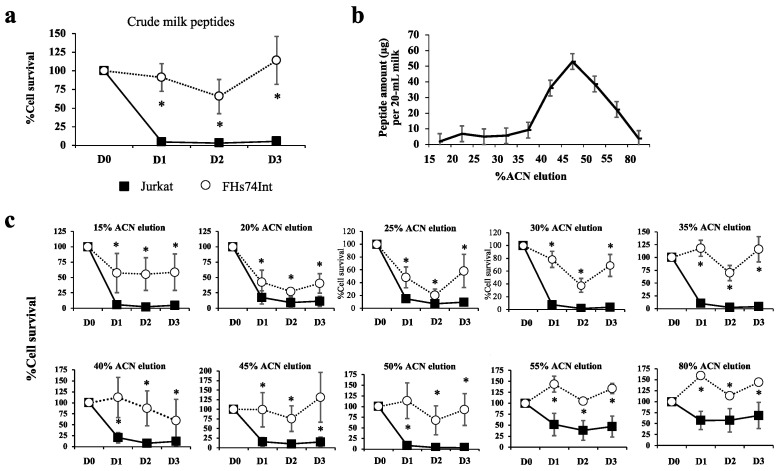
Fractionation of peptides from human milk and cytotoxicity of fractions towards leukemic and normal cells. Ten samples of human milk were divided into 3 pools. Twenty milliliters of each pool were centrifuged at 4 °C to remove cells, lipid, and extracellular vesicles. The crude milk peptides obtained from each pool were separately eluted through a cut-off column of <3 kDa, and the eluate was loaded to a C18 SPE column. Milk peptides bound to the C18 SPE column were eluted with various concentrations of acetonitrile (ACN) from 15% ACN to 80% ACN (1 mL each). Eluted fractions of milk peptides were dried using a SpeedVac concentrator, resuspended in a culture medium and then used to treat Jurkat (black square) and FHs74Int cells (white circle), using 3 biological replicates. WST-1 assay was applied to measure cell viability. (**a**) % cell survival (mean ± SEM) after the treatment of cells with crude milk peptides for 1 (D1), 2 (D2), or 3 (D3) days, compared to those of the untreated control (D0); (**b**) amounts of peptides eluted from the C18 SPE column eluted in a stepwise manner using 1 mL each of increasing ACN concentrations of 15%, 20%, 25%, 30%, 35%, 40%, 45%, 50%, 55%, and 80% ACN. Peptides were quantitated by the Bradford method and shown as mean ± SEM; (**c**) % cell survival (mean ± SEM) after the treatment of cells with eluates obtained at different %ACN concentration, for 1 day (D1), 2 days (D2), or 3 (D3) days, compared to those of the untreated controls (D0). * *p* < 0.05 comparing to the untreated condition.

**Figure 3 biomedicines-09-00981-f003:**
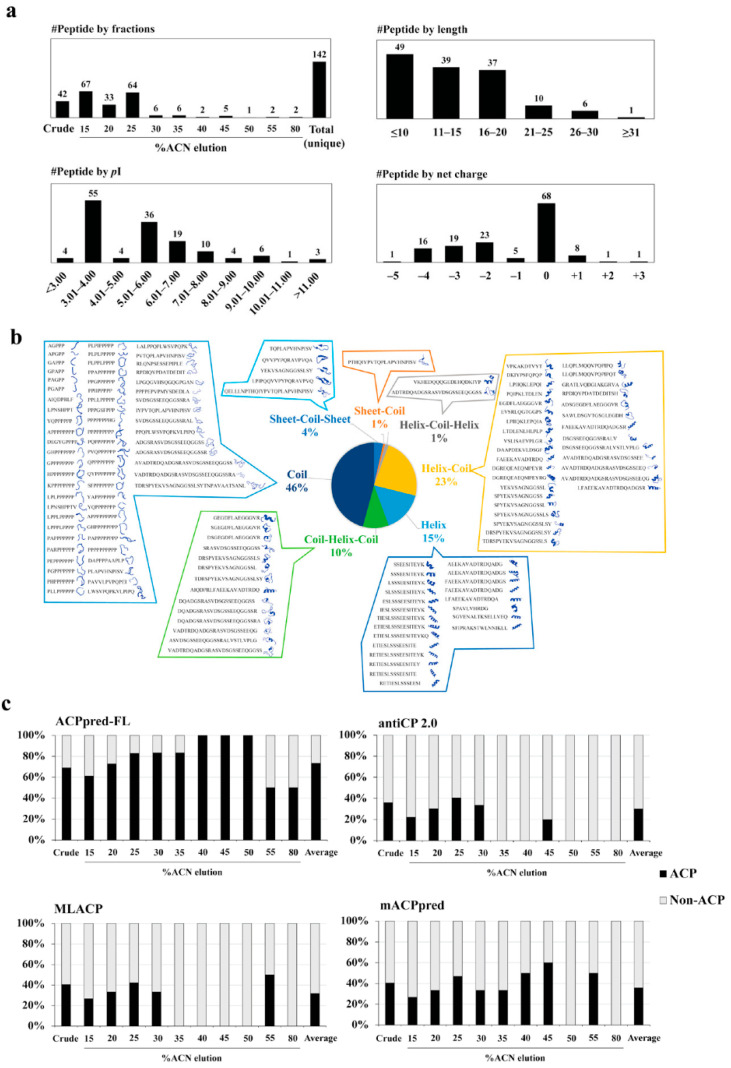
Predicted physicochemical, structural, and machine learning-based anticancer peptide screening of naturally occurring human milk peptides. (**a**) Distribution of the unique human milk peptides identified by LC-MS/MS; (**b**) the distribution and secondary structure of all identified peptides predicted by PEP-FOLD3 software; (**c**) proportions of predicted anticancer property (ACP) vs. non-ACP peptides using four ACP machine learning programs, including ACPpred-FL, antiCP 2.0, MLACP, and mACPpred. The percentages of ACP (black) and non-ACP (gray) peptides were calculated as: number of ACP or non-ACP predictions/number of total identified peptides in each individual fraction × 100%. Full results of in silico ACP screening are provided in Appendix A.

**Figure 4 biomedicines-09-00981-f004:**
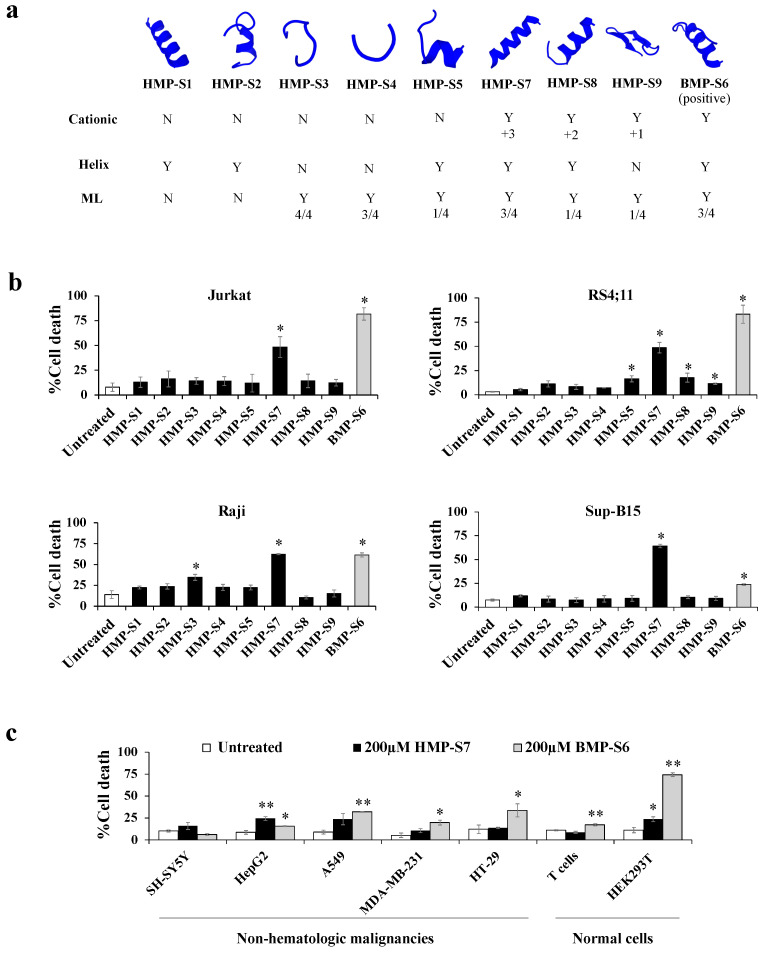
Effect of the 8 synthetic human milk peptides (HMPs) on leukemic and nonleukemic cell lines. (**a**) Eight HMPs and a positive control (BMP-S6) were selected and synthesized to test for antileukemic activity. The properties of these peptides are summarized, namely predicted secondary structure, cationic nature, helix content, machine learning (ML) prediction of ACP (details of the selected peptides are summarized in Appendix A); (**b**) four leukemic cell lines, namely Jurkat, Raji, RS4;11, and Sup-B15, were treated with the 8 synthetic HMPs and the control BMP-S6 at 200 µM, and % cell death was observed after 24 h treatment using the trypan blue exclusion assay under a light microscope. The % cell death was calculated as: number of death cells/total cell number × 100%. The percentages of cell death of all 4 leukemic cell lines after HMP-S7 treatment were significantly increased; (**c**) in addition to leukemic cells, HMP-S7 was also tested on nonhematological malignant cell lines, including neuroblastoma (SH-SY5Y), hepatoblastoma (HepG2), lung cancer (A549), triple-negative breast cancer (MDA-MB-231), colon cancer (HT-29), as well as on normal cells, such as T cells and HEK293T embryonic kidney cells. Statistical significance of differences in % cell death was calculated using three biological replicates. * *p* < 0.05, ** *p* < 0.01 compared to untreated cells.

**Figure 5 biomedicines-09-00981-f005:**
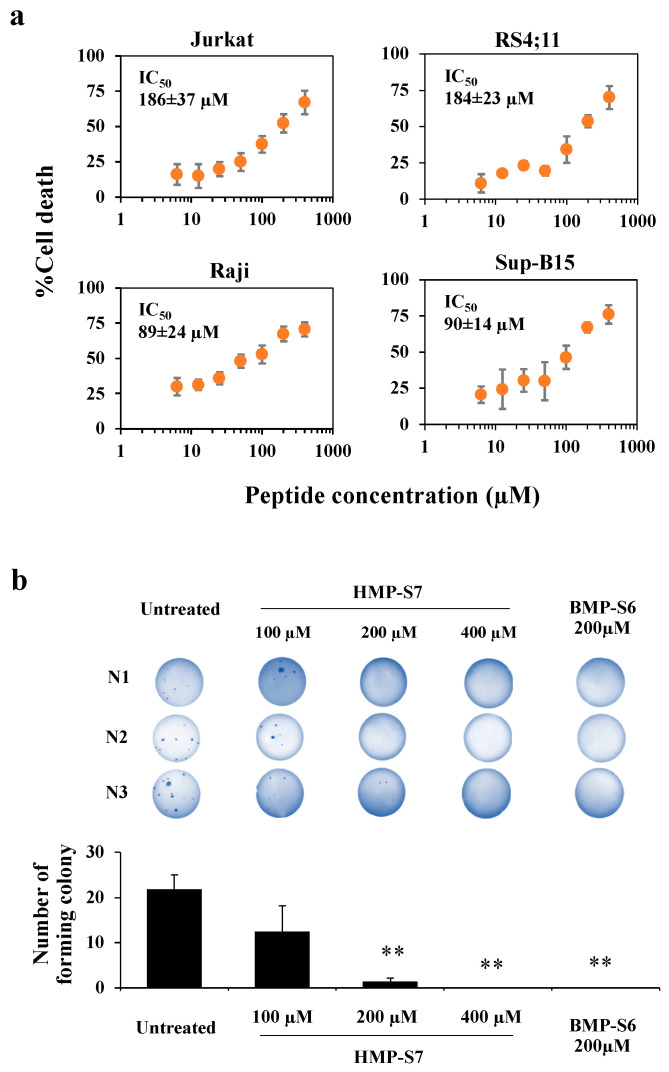
The inhibitory action of HMP-S7 on leukemic cells. (**a**) Four leukemic cell lines, namely Jurkat, Raji, RS4;11, and Sup-B15 cells, were treated with HMP-S7 at various concentrations (0–400 µM) for 24 h, and the % cell death (mean ± SD) was determined using trypan blue exclusion assay; (**b**) effects of HMP-S7 (100, 200, and 400 µM) and the positive control BMP-S6 (200 µM) on the colony forming ability of Jurkat cells (three independent experiments; N1–N3). After the treatment with the test peptide for 24 h, cell suspensions were allowed to form colonies in soft agar for 20 days. The colonies in the soft agar were stained with crystal violet and counted. ** *p* < 0.01 compared to untreated condition (*n* = 3 biological replicates).

**Figure 6 biomedicines-09-00981-f006:**
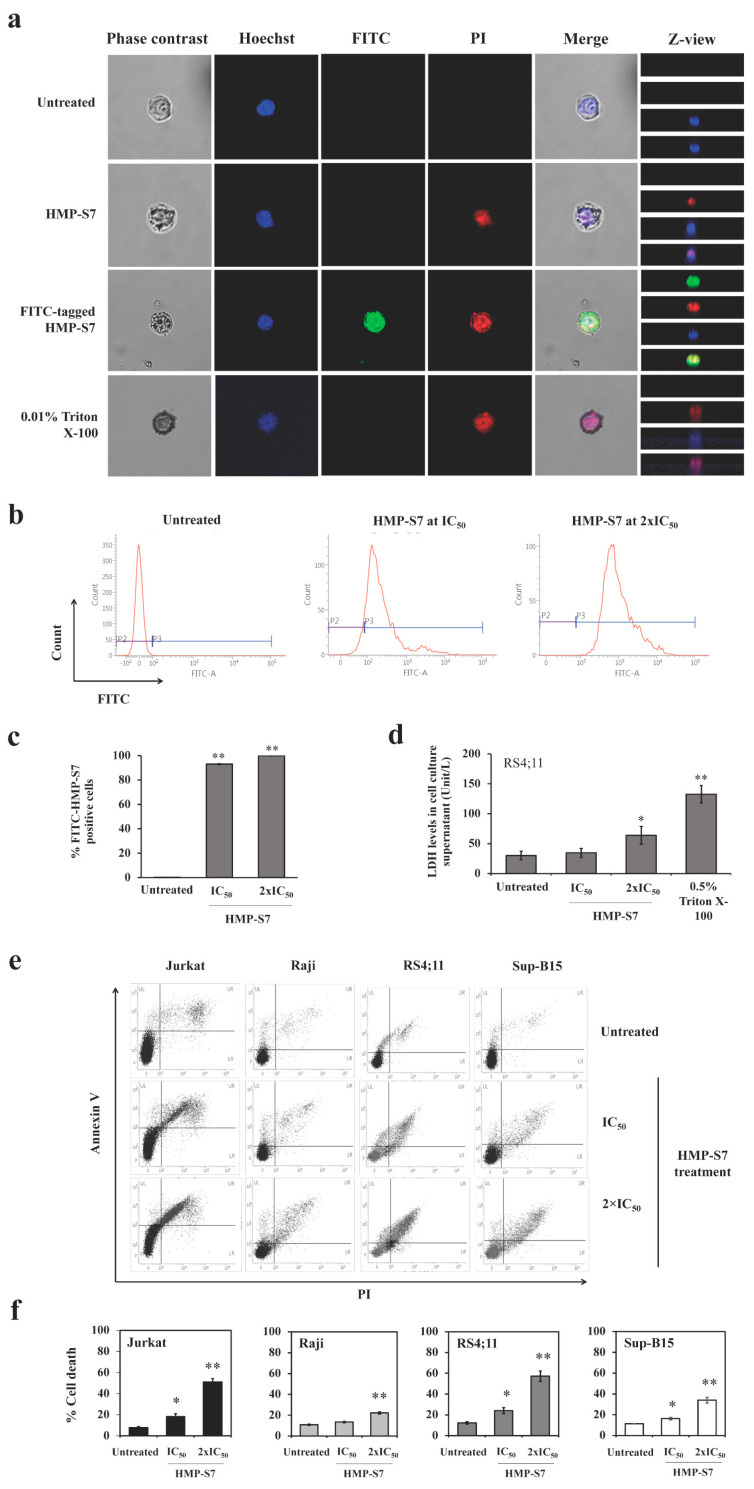
HMP-S7 action on internalization and leukemic cell death. Jurkat cells were treated with HMP-S7 conjugated with or without fluorescein isothiocyanate (FITC) at IC_50_ and 0.01% Triton X-100 (positive control) and stained with propidium iodide (PI) to observe membrane permeability. (**a**) The FITC-tagged HMP-S7 (at IC_50_ and 2 × IC_50_) was internalized into the cytoplasm of Jurkat cells; (**b**) flow cytometry of FITC-tagged HMP-S7-treated Jurkat cells; (**c**) flow cytometric results as %FITC-positive Jurkat cells; (**d**) lactate dehydrogenase (LDH) release assay showing the LDH level in the culture supernatant as the evidence of cellular membrane disruption of RS4;11 cells treated with HMP-S7 at IC_50_ and 2 × IC_50_ for 24 h. Triton X-100 treated cells were used as a positive control; (**e**) flow cytometric cell death assay using Annexin-V/PI co-staining. Four leukemic cell lines, namely Jurkat, Raji, RS4;11, and Sup-B15, were treated with HMP-S7 at IC_50_ and 2 × IC_50_ for 24 h; (**f**) % cell death composed of the upper left, upper right, and lower right quadrants (early and late apoptosis and necrosis) of flow cytometric data. * *p* < 0.05, ** *p* < 0.01 compared to untreated condition. All experiments were performed in three biological replicates.

**Figure 7 biomedicines-09-00981-f007:**
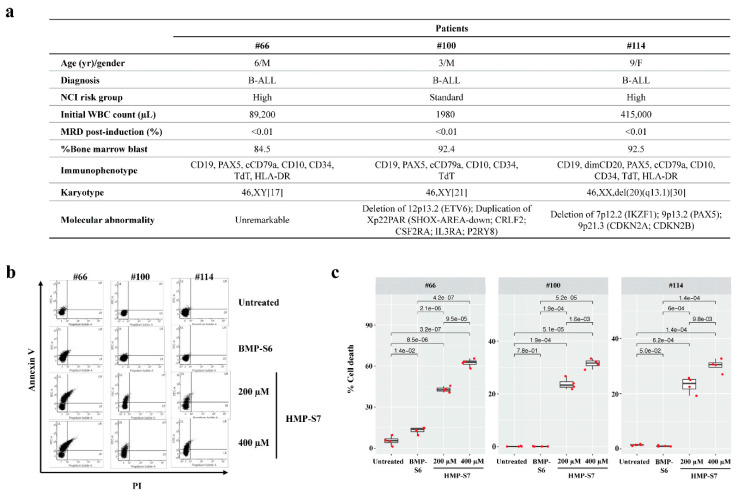
HMP-S7 induced patient-derived leukemic cell death ex vivo. Bone-marrow-derived lymphoblasts were collected from three leukemic patients and were processed as described in “Materials and Methods” section. (**a**) Demographic data of three leukemic patients. Patient-derived lymphoblasts were treated with two doses of HMP-S7 or with BMP-S6 (a positive control from previous experiments); (**b**) flow cytometric analysis with annexin V/PI co-staining at 72-h post-treatment. Patient-derived leukemic cells were treated with 200 and 400 µM HMP-S7, while 200 µM BMP-S6 was included for comparison; (**c**) % cell death including the upper left, upper right, and lower right quadrants of flow cytometric data (*n* = 4 replicates per condition). Abbreviations: B-ALL, B-cell acute lymphoblastic leukemia; F, female; M, male; MRD, minimal residual disease; NCI, National Cancer Institute; WBC, white blood cells.

## Data Availability

All data are available in the main text or the supplementary materials.

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
