# Peer review of "HMP-S7 Is a Novel Anti-Leukemic Peptide Discovered from Human Milk"

_biomedicines, 2021, doi:10.3390/biomedicines9080981_

Round 1
Reviewer 1 Report
A complete procedure for the discovery of novel peptides from human breast milk is presented, integrating iquid chromatography-tandem mass spectrometry, in silico screening using machine learning algorithms, etc.. Potential candidates were selected for functional studies to determine anti-leukemic activity. Taken together, a novel anti-leukemic peptide was identified from human milk. This anti-leukemic peptide is a cationic peptide with alpha-helical structure that selectively kills leukemic cell lines in vitro and exhibits its cytotoxicity against patient-derived leukemic cells ex vivo.
The manuscript is well writen, data are well presented and easy to follow. Morevoer, the limitations of the study are clarly presented. Therefore, my recomendation is accepting the manuscript in the present form.
Author Response
Reviewer#1
A complete procedure for the discovery of novel peptides from human breast milk is presented, integrating iquid chromatography-tandem mass spectrometry, in silico screening using machine learning algorithms, etc.. Potential candidates were selected for functional studies to determine anti-leukemic activity. Taken together, a novel anti-leukemic peptide was identified from human milk. This anti-leukemic peptide is a cationic peptide with alpha-helical structure that selectively kills leukemic cell lines in vitro and exhibits its cytotoxicity against patient-derived leukemic cells ex vivo.
The manuscript is well written, data are well presented and easy to follow. Morevoer, the limitations of the study are clearly presented. Therefore, my recommendation is accepting the manuscript in the present form.
We are grateful that the Reviewer found this manuscript is well written, well presented, and easy to follow. We also appreciate the recommendation of the Reviewer that the manuscript be accepted in the present form. Thank you for your support.
Reviewer 2 Report
In this study, the authors tried to use an integrative workflow combining mass spectrometric peptide library construction, in silico anticancer peptide screening, and in vitro leukemic cell studies to discover a novel anti-leukemic peptide HMP-S7 from human breast milk. The cytotoxic activity of HMP-S7 against four distinct leukemic cell lines was tested and its mechanism was studied. This study is quite interesting to researchers focusing on peptides cancers therapy, however, the following issues should be addressed prior to its acceptance for publication.
L85, “PGPIPN hexapeptide of bovine b-casein can inhibit invasion and migration of human ovarian cancer cells [23]. ACFP, an anti-cancer fusion peptide derived from bovine b-casein and lactoferrin, can inhibit viability and promote apoptosis in primary ovarian cancer cells”
The abbreviation should have a full-name.
L240, LDH assay, “Then 1 mL of supernatant was collected and LDH enzyme was measured from conversion of lactate to pyruvate and NADH, detected as absorbance at 340 nm using Abbott Architect C16000 clinical chemistry analyzer”
The assay should contain NAD and other chemicals. Please make more detail description to this assay.
L301, “the crude milk peptide fraction was tested to observe cytotoxic effects against Jurkat (T lymphoblastic leukemia) and FHs74Int cells (the representative normal intestinal epithelium), respectively”
The authors had to very careful address the toxicity to normal cell with the milk peptide fraction and specify the reasons of these milk peptides toxicity.
Fig. 3 “including ACPpred-FL, antiCP 2.0, MLACP, and mACPpred (Figure 3c). The results showed eight human milk peptides and one bovine milk peptide.”
How to screen the eight peptides from these graphs? The ACPpred-FL and MLACP had a quite different trend of ACP and non-ACP at 40-50%ACN elution.
Fig. 5(b) The N1-N3 should be claimed.
Fig 7(a) is a Table and the reviewer suggest it been subjected to supplementary material.
Fig 7(b) graph should be enlarged.
Fig 7c what is the scientific values in these graphs?
Author Response
Reviewer#2
In this study, the authors tried to use an integrative workflow combining mass spectrometric peptide library construction, in silico anticancer peptide screening, and in vitro leukemic cell studies to discover a novel anti-leukemic peptide HMP-S7 from human breast milk. The cytotoxic activity of HMP-S7 against four distinct leukemic cell lines was tested and its mechanism was studied. This study is quite interesting to researchers focusing on peptides cancers therapy, however, the following issues should be addressed prior to its acceptance for publication.
We glad that the Reviewer found this study to be interesting for researchers interested in anticancer peptide therapy. We appreciate the constructive comments and helpful suggestions of the Reviewer. We have made corrections as appropriate and feel that the quality of the manuscript has been significantly improved as a result.
1. L85, “PGPIPN hexapeptide of bovine b-casein can inhibit invasion and migration of human ovarian cancer cells [23]. ACFP, an anti-cancer fusion peptide derived from bovine b-casein and lactoferrin, can inhibit viability and promote apoptosis in primary ovarian cancer cells”
The abbreviation should have a full-name.
RESPONSE: Thank you for your suggestion. The full names of PGPIPN and ACFP have been added to the Introduction (page 2 lines 86-88).
2. L240, LDH assay, “Then 1 mL of supernatant was collected and LDH enzyme was measured from conversion of lactate to pyruvate and NADH, detected as absorbance at 340 nm using Abbott Architect C16000 clinical chemistry analyzer”
The assay should contain NAD and other chemicals. Please make more detail description to this assay.
RESPONSE: Thank you for pointing this out. More details have been provided for the LDH assay including NAD and other chemicals to the Materials and Methods (page 6, lines 247-252).
3. L301, “the crude milk peptide fraction was tested to observe cytotoxic effects against Jurkat (T lymphoblastic leukemia) and FHs74Int cells (the representative normal intestinal epithelium), respectively”
The authors had to very careful address the toxicity to normal cell with the milk peptide fraction and specify the reasons of these milk peptides toxicity.
RESPONSE: The authors appreciate this reviewer’s comment since using the term “to observe cytotoxic effects” for crude milk peptide fractions to normal cells may lead to misunderstanding and confusion. Therefore, the term “cytotoxic effects” is changed to “the effects on cell viability” (page 7, line 304).
4. Fig. 3 “including ACPpred-FL, antiCP 2.0, MLACP, and mACPpred (Figure 3c). The results showed eight human milk peptides and one bovine milk peptide.”
How to screen the eight peptides from these graphs?
RESPONSE: We apologize for the lack of clarity. Table S2 shows all the properties of each of the 142 peptides identified by mass spectrometry, including amino acid sequence and eluted fraction, together with predicted physicochemical, structural and anticancer properties. Thus, the 8 human milk peptides were selected based on a combination of properties shown in Table S2. Figure 3 shows the characteristics of the peptides in the crude milk fraction and eluted in different fractions of the C18 SPE column, with Figure 3a showing the distribution of peptides in different fractions, Figure 3b showing the predicted conformation, and Figure 3c showing the ACP or non-ACP predicted by four machine learning programs. Since, all the properties of each peptide are shown in Table S2, the words “Table S2” were added after Figure 3a, 3b and 3c in Results (page 8, line 349; and page 9, lines 353 and 360). Moreover, to improve understanding, the text has been modified in page 8, lines 336-337; and page 9, lines 360-365.
The ACPpred-FL and MLACP had a quite different trend of ACP and non-ACP at 40-50% ACN elution.
RESPONSE: During our initial study, we observed that the predictive performance of each ACP prediction software depends on the machine learning model implemented and the datasets used for training and testing. Thus, to improve the accuracy of ACP prediction, we applied four accessible ACP prediction software, and the positive results were counted for ranking. Since ACPpred-FL uses a sequence based predictor method and MLACP is built on the Random Forest model, it was not surprising that the two algorithms result in quite different trends of ACP prediction for peptides eluted at 40-50% ACN elution.
5. Fig. 5(b) The N1-N3 should be claimed.
RESPONSE: We thank the reviewer for this comment and have added this information into Figure 5 legend (page 12, line 419).
6. Fig 7(a) is a Table and the reviewer suggest it been subjected to supplementary material.
RESPONSE: We thank the reviewer for this constructive suggestion. The main objective of Figure 7 was to demonstrate the anti-leukemic effect of HMP-S7 against patient derived B-ALL with heterogeneity in demographic, clinical, immunophenotypic and genetic profiles. Since this patient information is rather complicated, but important in showing this heterogeneity in profile, the authors believe that presenting this information as Figure 7a together with Figure 7b and 7c will make it easier for the reader to appreciate the data.
7. Fig 7(b) graph should be enlarged.
RESPONSE: Thank you for this suggestion. Figure 7 has been enlarged to improve the overall resolution of this figure as suggested.
8. Fig 7c what is the scientific values in these graphs?
RESPONSE: The values in Figure 7c are p-values.
Round 2
Reviewer 2 Report
This MS is now acceptible for publication.